# Fast Bayesian Optimization of Function Networks with Partial Evaluations

Poompol Buathong[1]  Peter I. Frazier[2]

[1]Center for Applied Mathematics, Cornell University, Ithaca, NY, USA
[2]School of Operations Research and Information Engineering, Cornell University, Ithaca, NY, USA

**Abstract**  Bayesian optimization of function networks (BOFN) is a framework for optimizing expensive-to-evaluate objective functions structured as networks, where some nodes' outputs serve as inputs for others. Many real-world applications, such as manufacturing and drug discovery, involve function networks with additional properties – nodes that can be evaluated independently and incur varying costs. A recent BOFN variant, p-KGFN, leverages this structure and enables cost-aware partial evaluations, selectively querying only a subset of nodes at each iteration. p-KGFN reduces the number of expensive objective function evaluations needed but has a large computational overhead: choosing where to evaluate requires optimizing a nested Monte Carlo-based acquisition function for each node in the network. To address this, we propose an accelerated p-KGFN algorithm that reduces computational overhead with only a modest loss in query efficiency. Key to our approach is generation of node-specific candidate inputs for each node in the network via one inexpensive global Monte Carlo simulation. Numerical experiments show that our method maintains competitive query efficiency while achieving up to a 16× speedup over the original p-KGFN algorithm.

## 1 Introduction

Bayesian Optimization (BO) (Jones et al., 1998; Frazier, 2018) stands out as a robust and efficient method for solving optimization problems of the form $x^* \in \arg\max_{x \in \mathcal{X}} f(x)$, where the objective function $f(x)$ is a time-consuming-to-evaluate derivative-free black-box function.

Starting with an initial set of $n$ observations $D_n = \{(x_i, f(x_i)\}_{i=1}^n$, BO builds a surrogate model approximating the objective function $f(x)$. It then employs an acquisition function $\alpha_n(x)$, derived from the surrogate model, that quantifies the value of evaluating $f(x)$ at a new input point $x$. BO optimizes this acquisition function to choose the next input point $x$ at which to evaluate the objective function. Once this point is evaluated, the newly obtained data is incorporated into the observation set and the process iterates until an evaluation budget is exhausted.

The BO framework has demonstrated remarkable success across a broad spectrum of real-world applications, including hyperparameter optimization in machine learning (Snoek et al., 2012), materials design (Frazier and Wang, 2016), model calibration (Sha et al., 2020), agricultural planning (Cosenza et al., 2022), formulation design in food science (Khongkomolsakul et al., 2025) and manufacturing processes (Deneault et al., 2021).

While treating the objective function as a black box makes BO easy to apply, recently emerging grey-box BO methods (Astudillo and Frazier, 2021b) aim to accelerate optimization by exploiting side information produced during objective function evaluations and by modifying the objective function evaluation itself.

Bayesian optimization of function networks (BOFN) (Astudillo and Frazier, 2019, 2021a) is a leading grey-box BO approach. BOFN considers objective functions $f(x)$ that are compositions of two or more black-box functions, as in Figure 1. Such compositions are called *function networks* and are described with a directed acyclic graph (DAG) where each node in the graph is a function

and each edge is an input or output to/from a function. Function networks appear in real-world applications such as epidemic model calibration (Garnett, 2002), robotic control (Plappert et al., 2018) and solar cell production (Kusakawa et al., 2022). BOFN builds surrogates for these individual constituent functions by observing inputs and outputs (so-called intermediate outcomes) obtained during objective function evaluations. It then uses this extra information to guide the selection of points to evaluate, accelerating optimization.

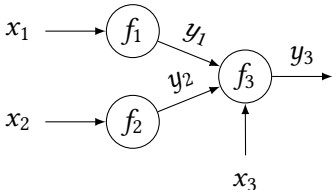

Figure 1: An example of an objective function modeled as a function network. The objective function's input is the vector $x \in \mathcal{X}$ comprised of three variables: $x_1$, $x_2$ and $x_3$. The objective is evaluated by evaluating individual functions $f_1$, $f_2$ and $f_3$ (shown as nodes in a directed acyclic graph) on their inputs (shown as edges in the graph). Its value is $f(x) = y_3(x) = f_3(f_1(x_1), f_2(x_2), x_3)$.

In the BOFN framework, Buathong et al. (2024) recently showed that optimization can be further accelerated by intelligently performing *partial evaluations*, i.e. evaluating only some of the functions in the function network in each iteration. In many applications, a partial evaluation is less time consuming than a full objective function evaluation and yet can provide high-value information. Using Figure 1 as an example, to optimize the final output $y_3$, one might evaluate $f_1$ at an input $x_1$ and observe $y_1 = f_1(x_1)$. This observation might suggest that $x_1$ is promising (if, for example, $y_1$ is large and our posterior suggests $f_3$ is increasing in $y_1$), in which case one might decide to evaluate $f_3$ at $y_1$ and some other previously-observed promising value for $y_2$. Or, if the observation suggests that $x_1$ is not promising, one might evaluate $f_1$ at a different input. Buathong et al. (2024) proposes an acquisition function, called the knowledge-gradient method for function networks with partial evaluations (p-KGFN), that guides the choice of individual functions to evaluate and the inputs at which to evaluate them by considering the value of the information obtained per unit evaluation cost.

While the p-KGFN acquisition function significantly reduces the time spent on objective function evaluation compared to previous BOFN and classical BO approaches, the acquisition function itself is time-consuming to evaluate and optimize. This makes the approach only useful in problems where the cost of objective function evaluation is so high that it outweighs the computational costs of acquisition function optimization, limiting its applicability. A key computational bottleneck is that computing p-KGFN requires solving a nested optimization problem with a Monte Carlo-based objective function over a mixed discrete/continuous search space: the set of nodes in the function network and their continuous vector-valued inputs.

To overcome this challenge and accelerate optimization across a much broader range of problems, we develop a fast-to-compute acquisition function, called Fast p-KGFN, that provides much of the benefit of p-KGFN at a fraction of the computational cost. Unlike the original p-KGFN, our approach avoids the need to solve a nested optimization problem. Instead, it uses a novel approach to identify a promising set of candidate measurements – candidate nodes and inputs at which to evaluate them. It then uses a novel approach for quickly evaluating the p-KGFN acquisition function for each candidate.

After introducing our new Fast p-KGFN approach, we validate the proposed algorithm performance across several test problems, demonstrating that it achieves comparable optimization results while significantly accelerating the original p-KGFN.

## 2 Problem Setting

We now formally describe function networks. Our presentation follows Astudillo and Frazier (2021a) and Buathong et al. (2024). We consider a sequence of functions $f_1, f_2, \ldots, f_K$ corresponding to nodes $\mathcal{V} = \{1, 2, \ldots, K\}$ in a DAG, $G = (\mathcal{V}, \mathcal{E})$. If an edge $(i, j)$ appears in the DAG $((i, j) \in \mathcal{E})$, this indicates that function $i$ produces output that is consumed as input by function $j$. For simplicity, we assume each $f_i$ produces scalar output though our approach generalizes easily to vector outputs.

Let $\mathcal{J}(k) = \{j : (j, k) \in E\}$ denote the set of parent nodes of node $k$, where $j$ is said to be a parent node of $k$ if $k$ consumes input produced as output from $j$. We assume that nodes are ordered such that $j < k$ for all $j \in \mathcal{J}(k)$. We suppose that there is an external input to the function network, indicated by $x$ and taking values in $\mathcal{X} \subseteq \mathbb{R}^d$. The output from the function network, and thus the objective function value, is determined by $x$. Let $\mathcal{I}(k) \subseteq \{1, \ldots, d\}$ be the set of components of $x$ that are taken as input by function $f_k$.

With these definitions, the output at node $f_k$ when the input to the function network is $x$ is given by a recursive formula:

$$y_k(x) = f_k(y_{\mathcal{J}(k)}(x), x_{\mathcal{I}(k)}), \quad \forall k = 1, \ldots, K, \tag{1}$$

where $y_{\mathcal{J}(k)}(x)$ denotes a vector of outputs from node $k$'s parent nodes, i.e. $y_{\mathcal{J}(k)}(x) = [y_j(x)]_{j \in \mathcal{J}(k)}$, and $x_{\mathcal{I}(k)} = [x_i]_{i \in \mathcal{I}(k)}$ are the external inputs to node $k$. We group the two types of inputs to $f_k$ – $y_{\mathcal{J}(k)}(x)$ and $x_{\mathcal{I}(k)}$ – into a single vector $z_k$. Then, the set of possible inputs to node $k$ is $\mathcal{Z}_k = \mathcal{Y}_{\mathcal{J}(k)} \times \mathcal{X}_{\mathcal{I}(k)}$ where $\mathcal{Y}_{\mathcal{J}(k)}$ represents the set of all possible values for the parent nodes' outputs and $\mathcal{X}_{\mathcal{I}(k)}$ denotes the set of possible values for $x_{\mathcal{I}(k)}$.

Our goal is to adaptively choose nodes $k$ and associated inputs $z_k$ to learn a near-optimal input $x$ to the function network that maximizes the output at the final node $y_K(x)$. For each node $k$, we assume an associated positive evaluation cost function $c_k(\cdot)$, and the learning task should be accomplished while minimizing the cumulative evaluation cost.

Buathong et al. (2024) allowed a restriction on the nodes and inputs evaluated that arises in some applications. In this restriction, when evaluating a function node $f_k$ at input that includes node output $y_{\mathcal{J}(k)}$, it is necessary to first provide parent node evaluations that produce this output. We do not consider this restriction here, though we believe that our approach can be extended to applications where this restriction holds. Instead, for each $k$, we assume that a set containing $\mathcal{Y}_{\mathcal{J}(k)}$ is known and that $f_k$ can be evaluated at any input in this set. This set containing $\mathcal{Y}_{\mathcal{J}(k)}$ could simply be $\mathbb{R}^{|\mathcal{J}(k)|}$ or it could be some strict subset. We argue that this setting is common. For example, in manufacturing, instead of producing an intermediate part with desired properties $y_{\mathcal{J}(k)}$ in-house, one can order the part from another supplier and combine it with other materials in a subsequent process. The external supplier might be too expensive to use during regular production but is acceptable during process development.

## 3 Existing Methods

We now present existing methods relevant to our method, focusing on Astudillo and Frazier (2021a) and Buathong et al. (2024). We first present the approach to inference proposed in Astudillo and Frazier (2021a) and used in Buathong et al. (2024), and which we also use. We then present two acquisition functions that we build on in our work.

**Inference**. To perform inference, Astudillo and Frazier (2021a) proposes to model each function $f_k$ with an independent Gaussian process (GP) (Williams and Rasmussen, 2006). This is tractable because our data is acquired as a collection of input/output pairs for each node $k$, $\mathcal{D}_{n_k(n),k} = \{(z_{i,k}, y_{i,k})\}_{i=1}^{n_k(n)}$, where $n_k(n)$ is the number observations at node $k$ after the total of $n$ evaluations. For simplicity, we will suppress the explicit dependency on $n$ and will use the notation $n_k$ throughout the manuscript. With data acquired in this way, the resulting posterior distributions on $f_1, \ldots, f_k$

**Algorithm 1** Proposed Fast p-KGFN Algorithm
___
**Input:**

The network DAG; Observation set $\mathcal{D}_n$; $c_k(\cdot)$, the evaluation cost function for node $k$, $k = 1, \ldots, K$; $B$, the total evaluation budget; $\mu_{n,k}$ and $\sigma_{n,k}$, the mean and standard deviation of the GP for node $k, k = 1, \ldots, K$ (fitted using initial observations $\mathcal{D}_n$);

**Output:** the point with the largest posterior mean at the final function node

1: $b \leftarrow 0$
2: **while** $b < B$ **do**
3:     Identify the maximizer of the current network posterior mean function $x_n^* \in$ $\arg\max_{x \in \mathcal{X}} v_n(x)$ and the current solution quality $v_n^*$;
4:     Generate an EIFN network candidate $\hat{x}_n$ by solving Eq. (2) with replacing $y_{n,K}^*$ by $v_n^*$;
5:     Sample a realization function $\hat{f}_k$ from a GP at node $k$, $\forall k = 1, \ldots, K$;
6:     Compute intermediate output $\hat{y}_k(\hat{x}_n)$ using a recursive formula in Eq. (1) and DAG structure;
7:     Construct a node-specific input $\hat{z}_{n,k} = (\hat{y}_{\mathcal{J}(k)}(\hat{x}_n), \hat{x}_{n,\mathcal{I}(k)})$, $\forall k = 1, \ldots, K$;
8:     Construct a discrete set $\mathcal{A}$ that includes points from the novel batch-Thompson sampling and local point methods and the current maximizer $x_n^*$;
9:     Evaluate a p-KGFN value, $\alpha_{n,k}(\hat{z}_{n,k})$, in Eq. (4) with using the discrete set $\mathcal{A}$, $\forall k = 1, \ldots, K$;
10:     Select $\hat{k} \leftarrow \arg\max_{k=1,\ldots,K} \alpha_{n,k}(\hat{z}_{n,k})$;
11:     Obtain the resulting evaluation $f_{\hat{k}}(\hat{z}_{n,\hat{k}})$;
12:     Update the GP model for node $\hat{k}$ with the additional observation $(\hat{z}_{n,\hat{k}}, f_{\hat{k}}(\hat{z}_{n,\hat{k}}))$;
13:     Update budget $b \leftarrow b + c_{\hat{k}}(\hat{z}_{n,\hat{k}})$ and the number of total evaluations $n \leftarrow n + 1$;
14: **end while**

**return** $\arg\max_{x \in \mathcal{X}} v_n(x)$ the maximum value of the posterior mean at the final node output.
___

remain conditionally independent GPs. We let $\mu_{n,k}(\cdot)$ and $\Sigma_{n,k}(\cdot, \cdot)$ denote the posterior mean and kernel associated with the GP on $f_k$ given $\mathcal{D}_{n_k,k}$ aften $n$ evaluations have been performed.

Let $\mathcal{D}_n = \cup_{k=1}^K \mathcal{D}_{n_k,k}$ be the combined observation set. The conditionally independent GP posterior distributions over $f_1, \ldots, f_K$ given $\mathcal{D}_n$ further induce a posterior distribution over the final node output $y_K(\cdot)$. However, due to its compositional network structure, this induced posterior distribution of $y_K(\cdot)$ is not Gaussian.

**The EIFN Acquisition Function.** Using the statistical model above, Astudillo and Frazier (2021a), proposed the EIFN acquisition function to select a candidate $\hat{x}_n \in \mathcal{X}$ at which to evaluate the entire function network. For example, this $\hat{x}_n$ in the Figure 1's example is a 3-dimensional input tuple $\hat{x}_n = (\hat{x}_{n,1}, \hat{x}_{n,2}, \hat{x}_{n,3})$. We refer to such evaluations of the entire function network as *full* evaluations, in contrast with our focus in this paper on partial evaluations. We introduce EIFN because we will use it as a tool in our approach.

To define the EIFN acquisition function, define $y_{n,K}^* = \max_{i=1,\ldots,n} y_K(x_i)$ as the current best observed value at the final node given $\mathcal{D}_n$. The EIFN at a proposed point $x$ is the expected improvement of $y_K(x)$ over the current $y_{n,K}^*$ under the current posterior. Specifically,

$$\text{EIFN}_n(x) = \mathbb{E}[(y_K(x) - y_{n,K}^*)^+ | \mathcal{D}_n], \tag{2}$$

where $(a)^+ = \max\{0, a\}$. The candidate selected is $\hat{x}_n \in \arg\max_{x \in \mathcal{X}} \text{EIFN}_n(x)$.

**The p-KGFN Acquisition Function.** Buathong et al. (2024) introduced the cost-aware *knowledge gradient for function networks with partial evaluations* (p-KGFN). This acquisition function proposes an individual candidate node $k$ and an associated input $\hat{z}_{n,k} \in \mathcal{Z}_k$ at each iteration $n$.

To define p-KGFN, let $v_n(x) = \mathbb{E}[y_K(x)|\mathcal{D}_n]$ be the posterior mean of the final node's output evaluated at network input $x$ and define the maximum value of this current posterior mean function

$$v_n^* = \max_{x \in \mathcal{X}} v_n(x), \tag{3}$$

as the current solution quality. At each BO iteration, p-KGFN loops through all function nodes and proposes a candidate $\hat{z}_{n,k}$ that solves: $\hat{z}_{n,k} \in \arg\max_{z_k \in \mathcal{Z}_k} \alpha_{n,k}(z_k)$, where

$$\alpha_{n,k}(z_k) = \frac{\mathbb{E}[\max_{x \in \mathcal{X}} v_{n+1}(x; z_k)|\mathcal{D}_n] - v_n^*}{c_k(z_k)} \tag{4}$$

and $v_{n+1}(x; z_k)$ denotes the new posterior mean of $y_K$ at $x$, conditioned on an unknown observation $y_k(z_k)$ evaluated at node $k$ with the input $z_k$, i.e. $v_{n+1}(x; z_k) = \mathbb{E}[y_K(x)|\mathcal{D}_n, y_k(z_k)]$. Interpreting $v_n^*$ and $\max_{x \in \mathcal{X}} v_{n+1}(x; z_k)$ as the expected quality of the best solution available before and after the observation of $y_k(z_k)$, the p-KGFN method proposes a node-specific candidate $\hat{z}_{n,k}$ that maximizes the expected increase in this measure of solution quality per unit evaluation cost. Then the p-KGFN compares these node-specific candidates and selects the node $\hat{k} \in \arg\max_{k=1,\dots,K} \alpha_{n,k}(\hat{z}_{n,k})$ to evaluate at its corresponding input $\hat{z}_{n,\hat{k}}$.

The p-KGFN acquisition function is challenging to optimize because it is defined by a complex nested expectation. Specifically, it is the expectation with respect to an unknown observation, $y_k(z_k)$, of the value of a maximization problem. This maximization problem is itself challenging to solve because its objective function $v_{n+1}(x; z_k)$ is also an expectation of the final node's value (under the updated posterior given a random observation $y_k(z_k)$). These expectations do not have analytical expressions and must be evaluated using Monte Carlo, quasi Monte Carlo, or numerical integration. Furthermore, generating candidates for each node requires solving separate p-KGFN optimization problems for each node, further compounding the computational challenges.

## 4 Our Method

With the challenge of optimizing the p-KGFN acquisition function in mind from the previous section, we introduce in this section a significantly faster-to-optimize acquisition function, Fast p-KGFN, that uses significantly less compute than the original p-KGFN while still providing a similar ability to find good solutions to the original optimization problem using a small number of low-cost partial evaluations.

This Fast p-KGFN consists of two key innovative components: (1) fast candidate generation that reduces a complex search over continuous node inputs to simply evaluating the acquisition function on a small discrete set; and (2) fast acquisition function computation, which enables efficient selection of the enumerated point from (1) with the best p-KGFN value. At each iteration, the method requires solving only one continuous optimization problem, and this continuous problem uses an objective function that is significantly more tractable than the original p-KGFN. Algorithm 1 outlines the complete procedure described in this section.

### 4.1 Fast Candidate Generation

To generate node-specific candidates as inputs for each node in the network, our Fast p-KGFN algorithm performs the following steps. In each iteration, given a DAG representing the function network and initial combined observation set $\mathcal{D}_n$, the proposed Fast p-KGFN first fits the the conditional posterior distributions of $f_1, \dots, f_K$, which together induce the posterior distribution for $y_K$. This is the same as the original p-KGFN.

Next, Fast p-KGFN optimizes a modified version of the EIFN acquisition function defined in Eq. (2). EIFN is modified by replacing $y_{n,K}^*$, the maximum function network output over the previously-evaluated network inputs, with $v_n^*$, the maximum conditional expected value of the

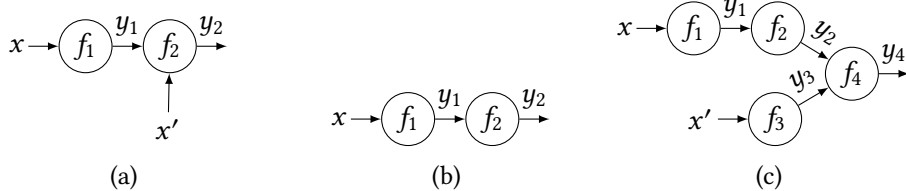

Figure 2: Function networks in the numerical experiments: (a) AckMat (b) FreeSolv and (c) Manu

function network's output over all possible network inputs. We use $v_n^*$ instead of $y_{n,K}^*$ because, during partial evaluations, the entire network is seldom fully evaluated with a single network input $x$. As such, $y_{n,K}^*$ is seldom updated. In contrast, $v_n^*$ is updated every time a new observation is evaluated. Optimizing this modified EIFN acquisition function produces a network candidate $\hat{x}_n$ containing values for all external inputs to the function network. For example, in Figure 1, optimizing EIFN gives a network candidate that takes the form $\hat{x}_n = (\hat{x}_{n,1}, \hat{x}_{n,2}, \hat{x}_{n,3})$.

Fast p-KGFN then draws a realization $\hat{f}_k$ of each function $f_k$ from its GP posterior and combines these sampled functions according to the DAG structure to construct a realization of all node outputs at the network candidate, $\hat{y}_k(\hat{x}_n)$. This is done using the recursive formula given by replacing $f_k$ by $\hat{f}_k$ and $y_k$ by $\hat{y}_k$ in Eq. (1). In Figure 1, this step yields the two intermediate outputs $\hat{y}_1(\hat{x}_n)$ and $\hat{y}_2(\hat{x}_n)$, corresponding to evaluations of the sampled function $\hat{f}_1$ at $\hat{x}_{n,1}$ and $\hat{f}_2$ at $\hat{x}_{n,2}$, respectively.

Finally, for each function node $k$, the algorithm constructs node-specific candidates $\hat{z}_{n,k} = (\hat{y}_{\mathcal{J}(k)}(\hat{x}_n), \hat{x}_{n,\mathcal{I}(k)})$ by concatenating the EIFN-generated candidate components $\hat{x}_{n,\mathcal{I}(k)}$ with the simulated intermediate outcomes from the parent nodes $\hat{y}_{\mathcal{J}(k)}(\hat{x}_n)$. For example, in Figure 1, node-specific candidates take the form $\hat{z}_{n,1} = \hat{x}_{n,1}$, $\hat{z}_{n,2} = \hat{x}_{n,2}$ and $\hat{z}_{n,3} = (\hat{y}_1(\hat{x}_n), \hat{y}_2(\hat{x}_n), \hat{x}_{n,3})$ at function nodes $f_1$, $f_2$ and $f_3$, respectively.

As we describe below in Section 4.3, the p-KGFN acquisition function will then be evaluated for each of these finitely many node-specific candidates to identify the one with the largest p-KGFN acquisition function value. To perform this evaluation quickly despite the fact that evaluation requires a nested expectation, we leverage a novel technique described below in Section 4.2.

## 4.2 Fast Acquisition Function Computation

This section describes an accelerated approach for computing the p-KGFN acquisition function, which is used as described below in Section 4.3.

Assuming that the maximizer $x_n^*$ of the current posterior mean $v_n^* = \max_{x \in \mathcal{X}} v_n(x)$ has been accurately identified, evaluating the p-KGFN acquisition function at a candidate node $k$ with input $z_k$ involves approximating the solution to the updated optimization problem: $\max_{x \in \mathcal{X}} v_{n+1}(x; z_k)$. Solving this problem is computationally expensive. Moreover, we must solve this optimization problem for many different sampled values of $y_k(z_k)$. (Each sampled value changes the function $v_{n+1}(x)$.) To solve these many related optimization problems quickly, we adopt an idea from the literature (Scott et al., 2011; Ungredda et al., 2022). Rather than optimizing each problem over the full continuous search space $\mathcal{X}$, our proposed discretization method optimizes them over a small finite discrete set $\mathcal{A}$ that is designed to include high-potential solutions.

Buathong et al. (2024) also leveraged this approach and proposed two approaches for constructing this discrete set $\mathcal{A}$: (1.) a Thompson sampling-based approach and (2.) local point sampling. We propose an improved method for constructing this set.

In the Thompson sampling-based method, $M$ function network realizations are sampled from the current posterior, and each realization is optimized independently to obtain a maximizer. The resulting $M$ maximizers are then included in $\mathcal{A}$. In this approach, increasing $M$, and thus the cardinality of $\mathcal{A}$, improves the accuracy of p-KGFN estimates but increases computational cost.

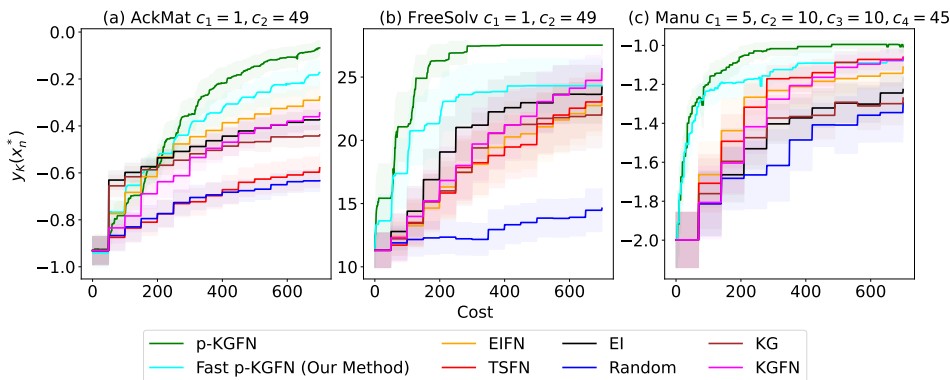

Figure 3: Optimization performance comparing between our proposed Fast p-KGFN algorithm and benchmarks including p-KGFN, EIFN, KGFN, TSFN, EI, KG and Random on three experiments: AckMat (left), Freesolv (middle) Manu (right).

Our goal here is to more intelligently select the set of points in $\mathcal{A}$ to achieve a better tradeoff between computational cost and the accuracy of p-KGFN estimation. Our novel strategy considers a large collection of $M$ posterior samples generated using Thompson sampling and selects a subset $\mathcal{S}_T$ of representative maximizers for $\mathcal{A}$. Specifically, our method proceeds as follows. First, our method samples $M$ function network realizations, each denoted as $\hat{f}_j^{\mathcal{A}}$ where the superscript $\mathcal{A}$ distinguishes these from the realizations used earlier in Section 4.1 for generating node-specific candidates. Then, our method selects a subset $\mathcal{S}_T$ of candidate points that perform well across the $M$ sampled network realizations. This is done by solving

$$\mathcal{S}_T \in \underset{\mathcal{S} \subset \mathcal{X}^{N_T}:|\mathcal{S}|=N_T}{\arg\max} \frac{1}{M} \sum_{j=1}^{M} \max_{x \in \mathcal{S}} \hat{f}_j^{\mathcal{A}}(x),$$

where the selected subset cardinality, $N_T = |\mathcal{S}_T|$, is a hyperparameter of the method. This approach can be viewed as a batch Thompson sampling method designed to generate a diverse set of high-potential candidates with significantly lower computational cost than optimizing each realization independently.

For local point sampling, we follow the random sampling procedure described in Buathong et al. (2024). A set $\mathcal{S}_L$ of local points with cardinality $N_L$ is generated around the current posterior mean maximizer $x_n^*$. A local point $x \in \mathcal{X}$ is generated by sampling uniformly among points satisfying $d(x, x_n^*) \leq r \max_{i=1,\ldots,d}(b_i - a_i)$, where $a_i$ and $b_i$ are the lower and upper bounds of the input of dimension $i^{th}$ and $r$ is a positive hyperparameter.

The final discrete set is $\mathcal{A} = \mathcal{S}_T \cup \mathcal{S}_L \cup \{x_n^*\}$ and is used in place of the original full search space $\mathcal{X}$ in solving updated posterior mean optimization problem, i.e. we solve $\max_{x \in \mathcal{A}} v_{n+1}(x; z_k)$.

### 4.3 Node Selection

After evaluating the p-KGFN candidates (nodes and input values) from Section 4.1 using the discretization approach from Section 4.2, the node $\hat{k}$ with the highest p-KGFN value is then selected for evaluation at the selected input $\hat{z}_{n,\hat{k}}$. The resulting data point $(\hat{z}_{n,\hat{k}}, f_{\hat{k}}(\hat{z}_{n,\hat{k}}))$ is added to the observation set $\mathcal{D}_n$. This process is repeated iteratively until the evaluation budget is depleted. The complete procedure is summarized in Algorithm 1.

## 5 Numerical Experiments

This section assesses the efficiency of the proposed Fast p-KGFN algorithm described in Section 4, with parameters $M = N_T = N_L = 10$ and $r = 0.1$ used in the new discretization method. To

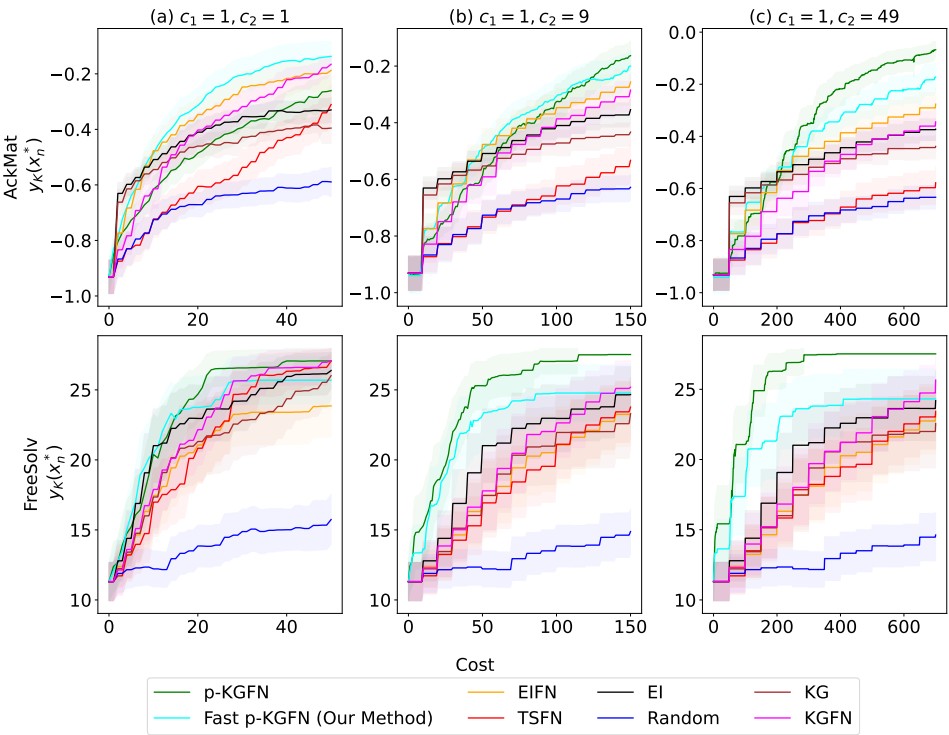

Figure 4: Cost sensitivity analysis for AckMat (top row) and FreeSolv (bottom row) problems with different costs (a) $c_1 = 1, c_2 = 1$; (b) $c_1 = 1, c_2 = 9$; and (c) $c_1 = 1, c_2 = 49$.

Table 1: Runtime comparison on 8-core CPUs for Fast p-KGFN, p-KGFN, and EIFN across AckMat, FreeSolv (three cost scenarios), and Manu. The table shows average runtime (with standard error) over 30 trials and BO courses with Fast p-KGFN speedup over p-KGFN shown in parentheses.

| Runtimes (mins per one BO iteration) | | | | |
|---|---|---|---|---|
| Problem | p-KGFN | EIFN | Fast p-KGFN (Our method) | (compared to p-KGFN) |
| (1.a) ActMat $c_1 = 1, c_2 = 1$ | $11.24 \pm 0.45$ | $2.81 \pm 0.32$ | $0.98 \pm 0.09$ | $(11.47\times)$ |
| (1.b) ActMat $c_1 = 1, c_2 = 9$ | $11.75 \pm 0.54$ | $0.78 \pm 0.08$ | $2.77 \pm 0.29$ | $(4.24\times)$ |
| (1.c) ActMat $c_1 = 1, c_2 = 49$ | $8.52 \pm 0.28$ | $0.69 \pm 0.04$ | $1.69 \pm 0.15$ | $(5.04\times)$ |
| (2.a) FreeSolv $c_1 = 1, c_2 = 1$ | $5.45 \pm 0.38$ | $0.53 \pm 0.03$ | $0.34 \pm 0.02$ | $(16.03\times)$ |
| (2.b) FreeSolv $c_1 = 1, c_2 = 9$ | $3.98 \pm 0.14$ | $1.70 \pm 0.19$ | $0.42 \pm 0.02$ | $(9.48\times)$ |
| (2.c) FreeSolv $c_1 = 1, c_2 = 49$ | $3.81 \pm 0.17$ | $1.00 \pm 0.09$ | $0.29 \pm 0.02$ | $(13.14\times)$ |
| (3) Manu $c_1 = 5, c_2 = 10, c_3 = 10, c_4 = 45$ | $7.80 \pm 0.43$ | $0.52 \pm 0.07$ | $1.40 \pm 0.10$ | $(5.57\times)$ |

show that our method provides competitive optimization performance, but offers significantly faster runtime than the original p-KGFN, we consider three test problems previously considered in Buathong et al. (2024): AckMat, FreeSolv and Manu, with structures presented in Figure 2.

AckMat is a synthetic two-node cascade network whose structure is commonly found in real-applications, such as multi-stage simulators and inventory problems. Here, we aim to find optimal values of $x$ and $x'$ which yield the highest value of $y_2$. FreeSolv (Mobley and Guthrie, 2014) is originally a materials design problem whose objective is to find an optimal small molecule $x$ whose negative experimental free energy $y_2$ is maximized with using a computational energy value $y_1$ as a pre-screen approximation of $y_2$. The similar network structure can be found in sequential processes. Manu is a problem representing multiple processes happening in the real manufacturing problems. Here, we want to identify the combinations of materials $x$ and $x'$ which yield the highest value of final product $y_4$.

We consider the cost scenarios: $c_1 = 1$ and $c_2 = 49$ for AckMat and FreeSolv problems, and $c_1 = 5$, $c_2 = 10$, $c_3 = 10$ and $c_4 = 45$ for Manu problem. The BO evaluation budget is set to 700. We defer the full descriptions of these problems to Appendix A.

We also perform additional experiments to assess the impact on Fast p-KGFN performance of each discrete point generation strategy described in Section 4.2, as well as to evaluate the influence of parameters $M$, $N_T$, $N_L$ and $r$ involved in discrete set construction. Both experiments are conducted on the AckMat problem with the $c_1 = 1$ and $c_2 = 49$ cost scenario. The complete results for these studies are presented in Appendices B and C, respectively.

## 5.1 Benchmarks and Comparison Metric

We evaluate the proposed Fast p-KGFN algorithm's optimization performance against several baseline methods, including standard Expected Improvement (EI) (Jones et al., 1998; Močkus, 1975), Knowledge Gradient (KG) (Frazier et al., 2008; Wu and Frazier, 2016), and simple random sampling (Random), which do not utilize network structure. Additionally, we compare to EIFN (Astudillo and Frazier, 2021a), Thompson Sampling for Function Networks (TSFN), and Knowledge Gradient for Function Networks (KGFN), which exploit network structure but require full evaluations. We also include the original p-KGFN algorithm, which supports partial evaluations and leverages network structure.

All algorithms start with the same uniformly sampled $2d + 1$ initial observations, fully evaluated across the network, where $d$ is the network dimension. Algorithms proceed until the budget of 700 is exhausted. Performance is averaged over 30 trials, each with different initial observations. We report at each iteration the average ground truth value $y_K(x_n^*)$, where $x_n^* \in \arg\max_{x \in \mathcal{X}} v_n(x)$, with confidence intervals, as in Buathong et al. (2024). Though EI, KG and Random do not leverage network structure in their sampling strategies, this metric is computed using a model that incorporates it. We also report the CPU runtimes of each algorithm to validate our claim that the proposed algorithm is faster than the original p-KGFN.

All algorithms are implemented using *BoTorch* (Balandat et al., 2020) in Python. Optimization settings, unless stated otherwise, follow the implementations in Buathong et al. (2024). Code to reproduce our numerical experiments is available at: https://github.com/frazier-lab/partial_kgfn.

## 6 Results

Figure 3 illustrates the performance of the proposed Fast p-KGFN algorithm compared to various benchmarks on the AckMat (left), FreeSolv (middle), and Manu (right) problems. Among the evaluated methods, all except p-KGFN and the proposed approach require full evaluations. Notably, only EIFN, KGFN, TSFN, p-KGFN, and the proposed method incorporate the network structure into their sampling strategies. Furthermore, both p-KGFN and the proposed algorithm support partial evaluations, enabling intermediate nodes to be evaluated at any input within their known ranges.

Overall, p-KGFN achieved the strongest optimization performance, closely followed by the proposed method. This result is expected, as p-KGFN fully optimizes the acquisition function to select the most promising node-specific candidate in each iteration. While the proposed method also supports partial evaluations, it reuses candidate points from EIFN, which may slightly degrade candidate quality and introduce a minor performance trade-off. Nonetheless, the proposed algorithm consistently outperforms the original EIFN, demonstrating the benefits of partial evaluations.

To further investigate the proposed method, we assess its sensitivity to varying evaluation costs. Specifically, we alter the cost of evaluating the second node in AckMat and FreeSolv, considering three scenarios: (a) $c_1 = c_2 = 1$ (BO budget = 50), (b) $c_1 = 1, c_2 = 9$ (BO budget = 150), and (c) $c_1 = 1, c_2 = 49$ (BO budget = 700). Figure 4 presents the results. They show that the proposed algorithm behaves similarly to p-KGFN, with the benefits of partial evaluations becoming more pronounced as the second node's evaluation cost increases. This allows p-KGFN and the proposed

method to find better solutions more efficiently than other competitors. These findings confirm that despite slightly lower-quality node-specific candidates, the proposed algorithm effectively leverages partial evaluations to achieve strong optimization performance.

Beyond optimization quality, the proposed method substantially improves computational efficiency over p-KGFN. Table 1 summarizes the average runtimes (across 30 trials on an 8-core CPU) for all cost settings, focusing on the proposed algorithm, EIFN, and p-KGFN. The proposed method consistently achieved lower runtimes, with the most notable speedup in FreeSolv under $c_1 = c_2 = 1$, where it attained a 16.03× improvement over p-KGFN. Appendix D provides a full comparison of the Pareto fronts over acquisition runtime and final objective value for all methods.

In our ablation study on discrete point generation strategies, we find that excluding the maximizer of the current posterior mean $x_n^*$ from the discrete set $\mathcal{A}$ significantly degrades the performance of Fast p-KGFN. The best results are achieved when using the full combination of batch Thompson points, local points, and the current maximizer, justifying the discrete set $\mathcal{A}$ used in our main experiments. Progress curves for this study are shown in Appendix B.

Fixing this full combination strategy, we further analyze the influence of parameters $M$ (number of network realizations), $N_T$ (number of batch Thompson points), $N_L$ (number of local points) and $r$ (positive parameter used to defined local points). Across a range of parameter values, Fast p-KGFN consistently maintains robust performance. Full results for this study are provided in Appendix C.

## 7 Conclusion

This work addresses Bayesian optimization of function networks with partial evaluations (p-KGFN), a framework designed for optimizing expensive objective functions structured as function networks, where each node can be queried independently with varying evaluation costs. While p-KGFN finds better solutions with lower evaluation budgets than traditional approaches, it suffers from high computational overhead due to its reliance on Monte Carlo simulations and the need to solve individual acquisition function problems for node-specific candidate selection in each iteration.

To overcome these challenges, we propose a faster variant of p-KGFN that leverages the expected improvement for function networks (EIFN) framework. Instead of solving separate acquisition problems for each node, the proposed method generates a single candidate for the entire network using EIFN and combines it with simulated intermediate outputs from the surrogate model to generate node-specific candidates. A cost-aware selection strategy then determines which node and corresponding input candidate to evaluate at each iteration.

We evaluate the efficiency of the proposed method across multiple test problems. Our approach achieves competitive optimization performance compared to p-KGFN while significantly reducing computational time, with the fastest runtime improvement exceeding 16×.

Despite these advantages, our method has some limitations. It assumes that intermediate nodes can be evaluated at any feasible input without requiring the outputs of parent nodes in advance, which may limit its applicability in more constrained function network scenarios. Extending the method to handle upstream-downstream dependencies is an important direction for future work.

## 8 Broader Impact Statement

After careful reflection and consideration, the authors believe that this work presents no notable negative impacts to the society or the environment.

### Acknowledgements

We would like to thank the anonymous reviewers for their comments. PB would like to thank the DPST scholarship project granted by IPST, Ministry of Education, Thailand for providing financial support. PF was supported by AFOSR FA9550-20-1-0351.

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

## A Test Problem Descriptions

### A.1 Problem 1: ActMat

We consider a two-node cascade network, as illustrated in Figure 2a. The first node is a Ackley function (Ackley, 2012) that takes a 6-dimensional input $x$ as input:

$$f_1(x) = -20 \exp\left(-0.2\sqrt{\frac{1}{6}\sum_{i=1}^{6} x_i^2}\right) - \exp\left(\frac{1}{6}\sum_{i=1}^{6}\cos(2\pi x_i)\right) + 20 + \exp(1),$$

Each dimension of $x$ lies in $[-2, 2]$. The second node processes the first node output denoted as $y_1$ with an additional parameter $x' \in [-10, 10]$ to produce the final output $y_2$. We use the negated Matyas function (Jamil and Yang, 2013) for this second node:

$$f_2(y_1, x') = -0.26(y_1^2 + x'^2) + 0.48y_1x',$$

We assume the range of $y_1 \in [0, 20]$. This bound is used by p-KGFN algorithm to generate candidates for the second node.

The evaluation costs for nodes are set as $c_1 = 1$ and $c_2 = 49$. The objective is to find $x$ and $x'$ that maximize the final output $y_2$.

### A.2 Problem 2: FreeSolv

FreeSolv is a molecular design test case built upon the available data set (Mobley and Guthrie, 2014), consisting of calculated and experimental hydration-free energies of 642 small molecules. The problem is constructed as a two-stage function network presented in Figure 2b. Here, the input $x \in [0, 1]^3$ is a continuous representation of small molecule extracted from a variational autoencoder model (Gómez-Bombarelli et al., 2018) and compressed by the principal component analysis. $f_1$ represents a mathematical model that takes input $x$ and is used to estimate the negative calculated free energy. The second node represents a wet-lab experiment that takes this calculated negative energy output as input and returns the negative experimental free energy, a target we aim to maximize. To construct this continuous network optimization problem, two GP models for calculated and experimental negative energies are separately fitted using all available data and the posterior mean functions of these two GP models are used as the functions $f_1$ and $f_2$, respectively.

We estimate the range of $y_1$ from the raw dataset and assume that $y_1 \in [-5, 30]$. Similar to AckMat problem, the evaluation costs are set to be $c_1 = 1$ and $c_2 = 49$.

### A.3 Problem 3: Manufacturing (Manu)

A manufacturing problem is constructed as the function network shown in Figure 2c. Each function is drawn from a GP prior with Matérn 5/2 kernels and varying length scale parameters to reflect the different complexities of the individual process. The length scale parameters are 0.631, 1, 1 and 3 for $f_1, f_2, f_3$ and $f_4$, respectively. The outputscale parameter is set to 0.631 for all functions, except $f_4$ which uses 10.

We assume the intermediate outputs' ranges as follows: $y_1 \in (-2, 2)$ and $y_2, y_3 \in (-1, 1)$,. The inputs $x$ and $x'$ are constrained to $(-1, 1)$. Evaluation costs are assigned as $c_1 = 5, c_2 = 10, c_3 = 10$ and $c_4 = 45$. The goal is to determine the optimal pair of raw materials $x$ and $x'$ that maximize the final output $y_4$.

## B Ablation Study: The Effects of Discrete Set Generation Methods

This section presents progress curves averaging over 30 trials of the Fast p-KGFN algorithm on the AckMat problem under the cost scenario $c_1 = 1, c_2 = 49$, using different configurations of the

discrete set $\mathcal{A}$ for approximating the p-KGFN acquisition function. Recall that $\mathcal{S}_T$ denotes the set of discrete points generated by the novel batch Thompson sampling method and its cardinality is denoted by $N_T$. $\mathcal{S}_L$ denotes the set of discrete points generated by local point sampling approach and its cardinality is denoted by $N_L$. $x_n^*$ denotes the maximizer of the current posterior mean function. We compare the following six configurations:

- Thompson + Local + Current Maximizer, the default setting used in the main numerical experiments: $\mathcal{A} = \mathcal{S}_T \cup \mathcal{S}_L \cup \{x_n^*\}$, with $N_T = N_L = 10$;

- Thompson + Local: $\mathcal{A} = \mathcal{S}_T \cup \mathcal{S}_L$, with $N_T = N_L = 10$;

- Thompson + Current Maximizer: $\mathcal{A} = \mathcal{S}_T \cup \{x_n^*\}$, with $N_T = 20$;

- Local + Current Maximizer: $\mathcal{A} = \mathcal{S}_L \cup \{x_n^*\}$, with $N_L = 20$;

- Thompson Only: $\mathcal{A} = \mathcal{S}_T$, with $N_T = 20$ and

- Local Only: $\mathcal{A} = \mathcal{S}_L$, with $N_L = 20$.

The results are shown in Figure 5. The default configuration achieves the best overall performance and is nearly matched by the Thompson + Current Maximizer strategy. Excluding the current maximizer $x_n^*$ dramatically reduces performance.

## C  Ablation Study: The Effects of Parameters in Batch Thompson Sampling and Local point Strategies for Discrete Set Construction

In this section, we perform ablation studies on the parameter settings used to construct the discrete set $\mathcal{A}$ in the Fast p-KGFN algorithm. Specifically, we vary the parameters $M$ (number of network realizations), $N_T$ (number of batch Thompson points), $N_L$ (number of local points), and the radius parameter $r$ that defines the neighborhood for local point sampling. All studies are conducted on the AckMat problem under the cost scenario $c_1 = 1$, $c_2 = 49$, using the best-performing strategy for constructing $\mathcal{A}$, which includes points from batch Thompson sampling, local point strategy, and the current posterior mean maximizer $x_n^*$.

We begin by fixing $r = 0.1$ and varying the values of $M$, $N_T$, and $N_L$. We compare the default setting ($M = N_T = N_L = 10$), used in the main experiments, against four alternatives: (1) $M = N_T = N_L = 5$; (2) $M = N_T = 10$, $N_L = 5$; (3) $M = N_L = 10$, $N_T = 5$; and (4) $M = N_T = N_L = 15$. As shown in Figure 6, the Fast p-KGFN algorithm maintains strong and consistent performance across all these configurations.

Next, we fix the default values $M = N_T = N_L = 10$ and vary the radius parameter $r$, considering $r = 0.01, 0.1$ (default), and $0.5$. The results in Figure 7 demonstrate that Fast p-KGFN is also robust to changes in the radius parameter.

## D  Pareto Front Comparison

Figure 8 presents a comprehensive Pareto front comparison averaging over 30 trials with different initial observations of all 8 methods across the 3 problems — 3 cost scenarios for AckMat, 3 for FreeSolv, and 1 for Manu — evaluated based on acquisition function optimization runtime and the objective function value at the best design found by the end of the optimization process. In nearly all cases, the original p-KGFN achieved the best objective value, while our Fast p-KGFN produced highly competitive solutions with significantly lower runtime.

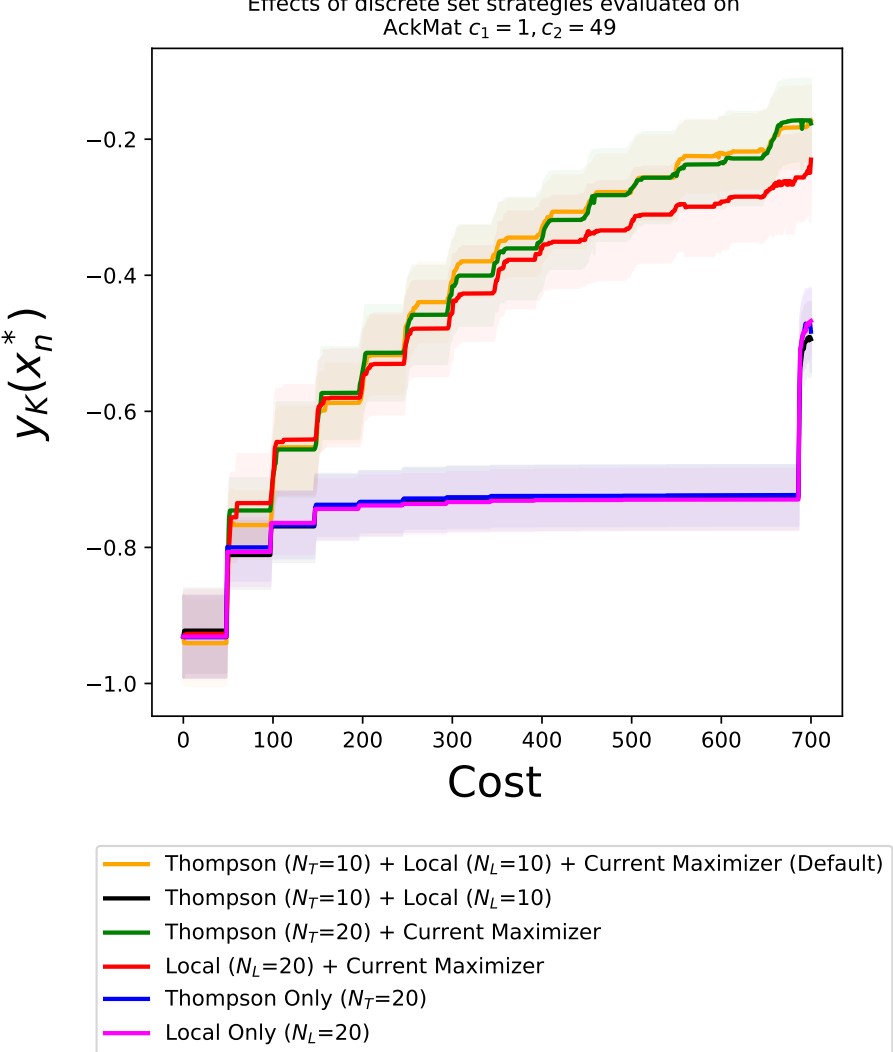

Figure 5: Ablation study focused on the discrete set generation method. Fast p-KGFN average performance over 30 trials on the AckMat problem ($c_1 = 1$, $c_2 = 49$), evaluated using six discrete set construction strategies combining batch Thompson sampling, local search, and the current posterior mean maximizer. The default configuration achieves the best performance compared to other considered configurations. Moreover, excluding the current maximizer from the discrete set significantly degrades the performance of the Fast p-KGFN algorithm.

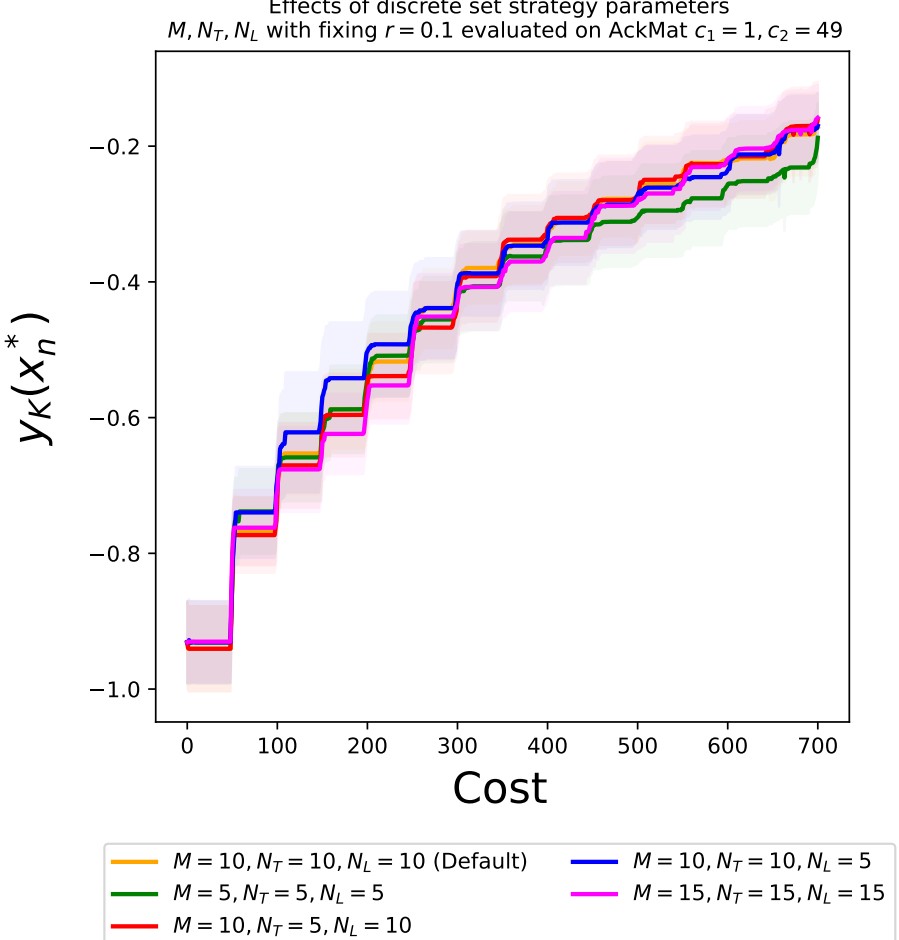

Figure 6: Ablation study focused on the discrete set generation parameters $M$, $N_T$ and $N_L$. Fast p-KGFN average performance over 30 trials on the AckMat problem ($c_1 = 1$, $c_2 = 49$) using fixed $r = 0.1$ and five configurations of discrete set parameters $M$, $N_T$, and $N_L$. Fast p-KGFN consistently maintains robust performances across all considered sets of these parameters.

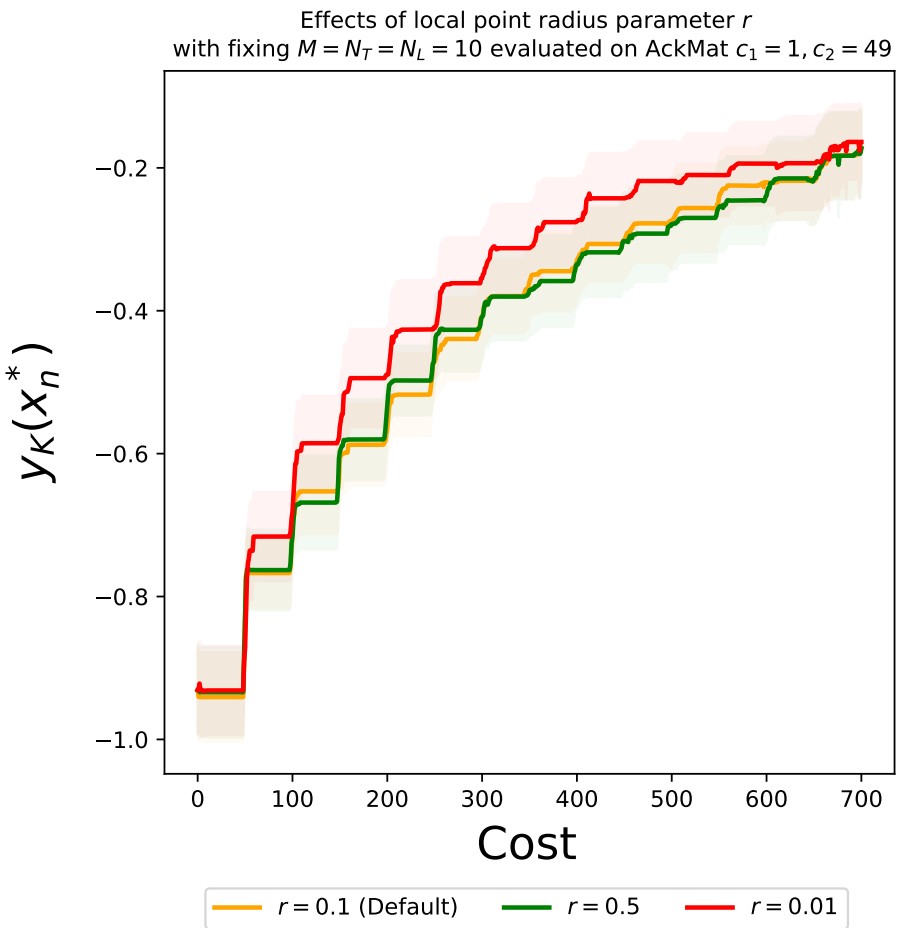

Figure 7: Ablation study focused on the discrete set generation parameter $r$. Fast p-KGFN average performance over 30 trials on the AckMat problem ($c_1 = 1$, $c_2 = 49$) with $M = N_T = N_L = 10$ and varying local radius $r = 0.01$, $0.1$ (default), $0.5$ for discrete set construction. The performance of Fast p-KGFN remains consistent across all considered values of $r$.

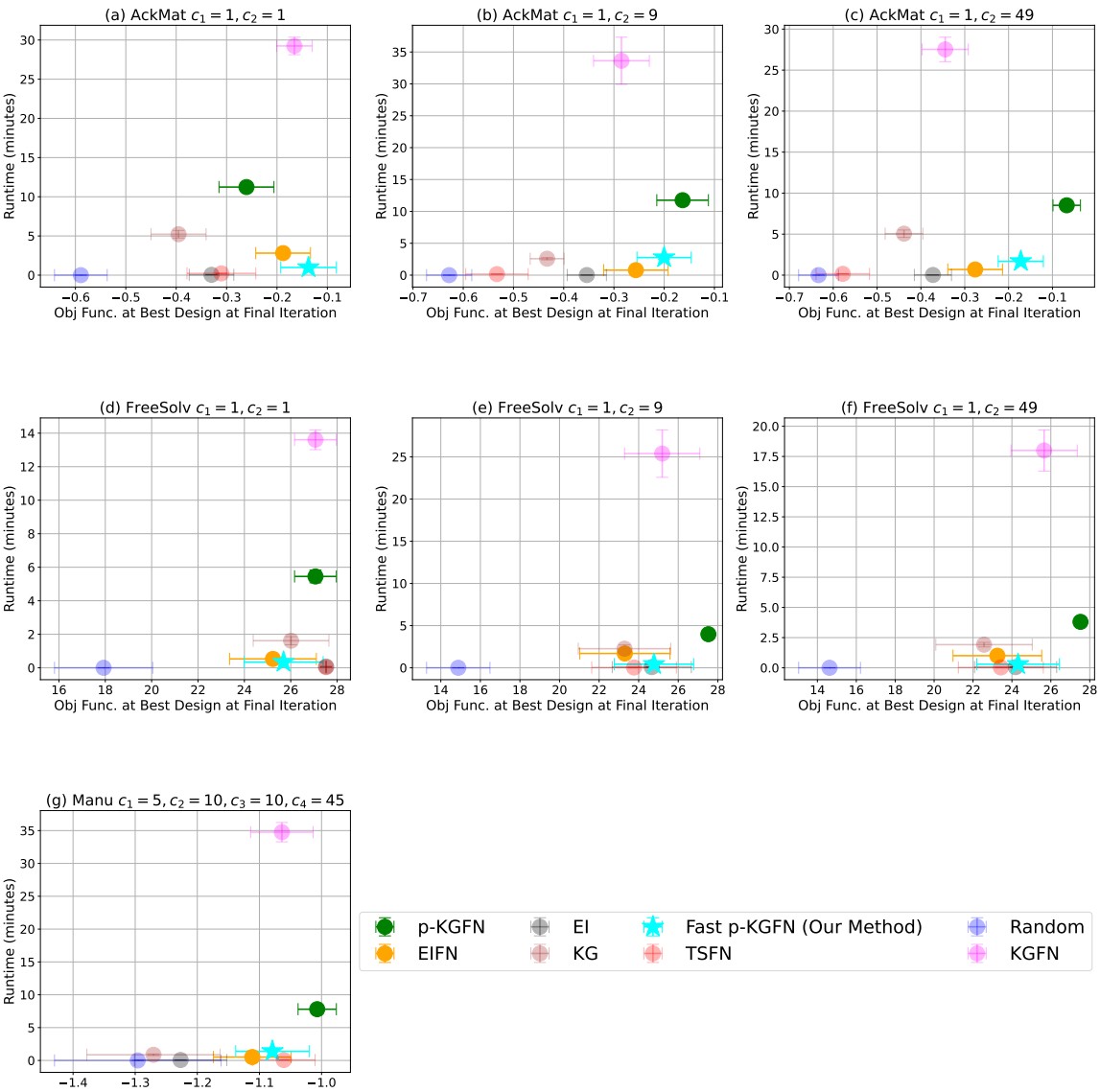

Figure 8: Pareto front comparison (acquisition optimization runtime vs. final objective value) averaged over 30 trials with error bars, comparing Fast p-KGFN against 8 baselines across 7 problems: AckMat (3 cost scenarios), FreeSolv (3 cost scenarios), and Manu (1 cost scenario). KGFN and p-KGFN achieve better (largest) final objective function values than previous methods in 6 of 7 problems but have much larger runtimes. Our method, Fast p-KGFN, achieves comparable solution quality (final objective value) while offering much lower runtimes than KGFN and p-KGFN.

