# OpenReview forum: "Fast Bayesian Optimization of Function Networks with Partial Evaluations"
_automl.cc/AutoML/2025/Methods_Track — AutoML 2025 Methods Track_

### Official Review · Reviewer_DWDz · 2025-04-25

**Comments To Authors:**

The paper presents Fast p-KGFN,  a computationally efficient algorithm for Bayesian Optimization of Function Networks (BOFN) that supports partial evaluations. This work addresses the computational inefficiencies in the existing p-KGFN algorithm, a recent method that, while effective, suffers from high computational costs due to its nested optimization structure and use of Monte Carlo methods. To do so, this paper introduces a method that generates node-specific candidates and efficiently computes acquisition function values using discretization technique.
- The authors introduce a batch Thompson sampling technique that produces a diverse set of potential solutions more efficiently.
- The experimental evaluation tests the proposed method across 3 benchmark problems with different cost scenarios. The results demonstrate the computational speedup (5-16×) while maintaining competitive optimization performance and a loss compared to vanilla p-KGFN.
However, this proposal has some weaknesses:
- The proposed method improves the computational speed of the vanilla p-KGFN. Although performance is close to p-KGFN, there is a tradeoff in quality, especially when full optimization is critical.
- The paper assumes that intermediate node evaluations are always possible, which may not hold in constrained real-world settings.
- Lack of formal theoretical guarantees or bounds on the proposed method's convergence properties.
- While the paper uses fixed parameters (M = N_T = N_L = 10 and r = 0.1) for the discretization method, there is no analysis of how sensitive the results are to (and why) these parameter choices.
- Although the authors tested on three different problems (AckMat, FreeSolv, and Manu), expanding the evaluation to more complex networks or real-world applications would strengthen the claims about the method's generalizability.
Some typos and mistakes, examples:
- line 129: For example, this … Figure 1 example.
- line 144: the the expected.
- line 19: "achieving up to a 16× speedup over existing BOFN methods" - this is slightly misleading as the 16× speedup is specifically over p-KGFN, not all BOFN methods (as shown in Table 1).

**Review Confidence:**

4

**Review Rating:**

7

---

### Official Review · Reviewer_xELG · 2025-04-30

**Comments To Authors:**

This paper introduces a new acquisition function (and a technique for its optimization) for Bayesian optimization (BO) on function networks. BO on function networks is a gray-box optimization technique where one observes the final output of a single black-box function and the outputs of upstream functions that serve as input to downstream functions. Each function can then be modeled with a surrogate model. Earlier acquisition functions (AFs) focused on finding inputs to all functions that give maximum utility of the final function. More recently, partial evaluations were considered where only some of the functions are observed, based on the assumption that a full observation is more costly. This AF, p-KFGN, however requires expensive Monte-Carlo simulations to estimate several expectations and therefore is relatively slow. This paper introduces several heuristics to speed up p-KFGN.

Strengths:
- While relatively new, the topics of BO on function networks and partial evaluations are relevant for several applications.
- The proposed heuristics are motivated reasonably.
- The proposed method is considerably faster than 'traditional' p-KFGN while inheriting many of its benefits.

Weaknesses:
- For most problems, the method does not give state-of-the-art performance. This is something the authors are very honest about, and the main strength is clearly the improved speed-up.
- There is no theoretical analysis of the error introduced by the heuristic design choices made by the authors. All analyses are purely empirical.
- Some parts of Section 4.2. are extremely confusing. It is unclear what the set $S$ is and why it is a true subset of $\mathcal{X}^{N_T}$. What is its cardinality $|\mathcal{S}|$. What is $M$ and why is it chosen larger than $N_T$? Where does $N_T$ come from, and what is $\mathcal{S}_{N_T}$ used for? Later, the paper only speaks about $\mathcal{S}_T$.  I strongly suggest to revise this part thoroughly.
- Both, Sections 4.1. and 4.2. would benefit from conceptual figures.

Minor comments:
- line 171: What does the 'these' in 'these steps' refer to? The following steps?
- line 188: 'despite the fact that *the* evaluation...'

**Review Confidence:**

3

**Review Rating:**

7

---

### Official Review · Reviewer_E2ms · 2025-04-30

**Comments To Authors:**

This paper presents an efficient approximation for a previously-proposed acquisition function for Bayesian optimization of function networks. Specifically, the authors develop an algorithm that approximately computes the knowledge gradient with partial evaluations: first, they compute the external inputs that optimize EIFN, the expected improvement acquisition function for function networks, $\hat{\vec{x}}$. They then use these external inputs and the GP beliefs on each function in the function network to generate intermediate outputs, which are then combined with the relevant elements of $\hat{\vec{x}}$ to get a set of $K$ candidate observations, one for each node in the function network. Finally, for each candidate observation, they approximate the knowledge gradient acquisition function by optimizing the posterior mean over a discrete set of observations (as opposed to the exact, continuous optimization); the discrete set is generated using a combination of Thompson samples and a local sampling method that targets locations near the current posterior maximizer.

Strengths:
- For the most part, I found the paper well-written, easy-to-follow and clear
- The proposed algorithm is technically sound and intuitive
- The empirical results are compelling and clearly demonstrate that the method significantly speeds up the original p-KGFN acquisition function

Weaknesses:
- While strong, I did find the empirical results a bit difficult to parse/lacking in certain aspects. In particular, given that computational speed is a relevant consideration, it would have been nice to see some sort of Pareto front between *all* the compared methods. It appears that Fast p-KGFN tends to outperform many of the other methods (although not necessarily at significance); however, the runtime is only compared against p-KGFN and EIFN, which neglects the fact that KGFN and TSFN appear to perform competitively with Fast p-KGFN (at least on two of the three benchmarks): if those methods are substantially faster than Fast p-KGFN, that would be relevant for a complete evaluation of the proposed method.
- The construction of the discrete set $\mathcal{A}$ in Section 4.2 was a bit arcane: it was not immediately clear to me why Thompson samples and local samples were combined to form the final search space as opposed to just using one method or the other. It would have been nice to see some sort of ablation study to compare the performance using just one set vs. the other vs. their union.
   - Unrelatedly, the notation in Section 4.2 leaves a lot to be desired: at various points $$S_{N_T}, \mathcal{S}_{N_T} \textrm{ and } \mathcal{S}_{T}$$ are all used to represent the same entity (I believe?). Also, in the equation for $\mathcal{S}_{N_T}$, shouldn't the $\in$ just be $=$? If not, how is this subsetting being done?
- I have some concerns about the relevance/impact of the paper: optimization of functions that follow the described function network structure already seems a bit niche but beyond even that, the authors acknowledge that their method is only applicable to settings where partial function observations are allowed and they are not applying the restriction considered by Buathong et al. (2024) that functions need to be provided with inputs that generate some desired intermediate output. This seems like a very realistic and relevant limitation that would have likely made their work more relevant. In general, it was not immediately obvious to me what applications would fall under the setting considered by this work; this concern was further exacerbated by the fact that all the experiments used synthetic or incredibly simplistic function networks.

Ultimately, I would characterize this work as a marginal advance over the prior work with a potentially limited impact. The algorithmic techniques used to speed-up the computation of the acquisition function do not provide a deeper insight into the workings of the method and heavily rely on relatively simple methods. That being said, I believe there exists a version of this work that warrants acceptance, one that makes a clear case for its applications, either through real-world benchmarks or more complex function networks.

**Review Confidence:**

4

**Review Rating:**

5

---

### Meta-Review · Area_Chair_AYPe · 2025-05-13

**Recommendation:** Accept
**Confidence:** 4

**Metareview:**

This paper considers a computational acceleration to p-KGFN, an acquisition strategy for Bayesian optimization in the setting where the obejctive function is "grey box" and known to decompose into a known function network, and partial evaluations of the function network are possible.

Most reviewers agree that the authors achieved what they set out to--a computation speed up of p-KGFN--and the results were reasonable. The two main criticisms levied at the paper were (1) the relatively narrow scope of results here (i.e., the authors' *only* consider this specific FN with partial evaluations setting) and (2) concerns about the comparison to vanilla p-KGFN and the trade-offs involved. Compounding this are concerns

My inclination is to agree with the reviewers in favor of acceptance despite these flaws for a number of reasons. First, I think the comparison to p-KGFN is ultimately pretty favorable here. The authors are explicitly targeting computational speed and not optimization performance here throughout the entire paper, and their method does appear to achieve results that are quite close. At a minimum, if vanilla p-KGFN is removed, their method does appear to get state of the art results. Most compelling in favor of the author is simplicity, which--while raised as a criticism--I think is actually a benefit, as the practical long lasting artifact from this paper is likely to be an extra keyword argument for a p-KGFN implementation with a docstring indicating this sets a mild or null performance trade-off for significant extra speed.

Second, I think the narrow focus is fine, given that the narrow focus of partial observability of the FN has already been litigated in prior work, and accepted in that work. In other words, since this problem setting already exists, it doesn't seem reasonable to litigate its value here (and my personal opinion is that it's reasonable and does exist in practice).